

# Estimating dune erosion at the regional scale using a meta-model based on Neural Networks

Panagiotis Athanasiou[1,2*], Ap van Dongeren[1,3], Alessio Giardino[4], Michalis Vousdoukas[5], Jose A.A. Antolinez[6] and Roshanka Ranasinghe[1,2,3]

[1]Deltares, Delft, Netherlands
[2]Water Engineering and Management, Faculty of Engineering Technology, University of Twente, Enschede, Netherlands
[3]Department of Coastal and Urban Risk & Resilience, IHE Delft Institute for Water Education, Delft, Netherlands
[4]Water Sector Group, Sustainable Development and Climate Change Department, Asian Development Bank, Manila, Philippines
[5]Joint Research Centre (JRC), European Commission, Ispra, Italy
[6]Department of Hydraulic Engineering, Faculty of Civil Engineering and Geosciences, Delft University of Technology, Delft, Netherlands

*Correspondence to*: Panagiotis Athanasiou (Panos.Athanasiou@deltares.nl)

**Abstract.** Sandy beaches and dune systems have high recreational and ecological value, while they offer protection against flooding during storms. At the same time, these systems are very vulnerable to storm impacts. Process-based numerical models are presently used to assess the morphological changes of dune and beach systems during storms. However, such models come with high computational costs, hindering their use in real-life applications which demand many simulations and/or involve a large spatial/temporal domain. Here we design a novel meta-model to predict dune erosion volume (DEV) at the Dutch coast,

based on Artificial Neural Networks (ANN), trained with cases from process-based modelling. First, we reduce an initial database of ~1,400 observed sandy profiles along the Dutch coastline to 100 representative Typological Coastal Profiles (TCPs). Next, we synthesize a set of plausible extreme storm events, which reproduces the probability distributions and statistical dependencies of offshore wave and water level records. We choose 100 of these events to simulate the dune response of the 100 TCPs using the process-based model *XBeach*, resulting in 10,000 cases. Using these cases as training data, we

design a 2-phase meta model, comprised of a classifying ANN (which predicts the occurrence (or not) of erosion) and a regression ANN (which gives a DEV prediction). Validation against a benchmark dataset created with *XBeach* and a sparse set of available dune erosion observations, shows high prediction skill with a skill score of 0.82. The meta-model can predict post-storm DEV $10^3$-$10^4$ times faster (depending on the duration of the storm) than running *XBeach*. Hence, this model may be integrated in early-warning systems or allow coastal engineers and managers to upscale storm forcing-dune response

investigations to large coastal areas with relative ease.



## 1    Introduction

Extreme wave and storm-surge conditions induced by tropical or extra-tropical storms can modify the morphological shape of sandy coasts and dunes dramatically (Vellinga, 1982; McCall et al., 2010; Castelle et al., 2015), with high impacts on local assets, infrastructure, ecosystems or touristic value. More importantly, at dune-protected coasts, the partial or total erosion of

the dunes during extreme storms can cause flooding of the hinterland (van Dongeren et al., 2018; Almar et al., 2021). The amount of dune erosion is associated with the offshore storm intensity (e.g., maximum wave height and storm surge), but can locally vary for the same storm, due to spatial variabilities of pre-storm foreshore and dune morphology and local hydrodynamics (Houser et al., 2008; Athanasiou et al., 2018; Beuzen et al., 2019). Accurate predictions of dune erosion during storms can be critical in early warning systems for the identification of dune erosion hotspots along long coastal stretches, as

well as for long-term coastal zone management, aiding in decision making and adaptation strategies to reinforce and/or nourish coasts and dunes.

Conceptual models that identify storm-impact regimes based on simple indicators using the relative elevations between the beach/dune and the total water levels have been previously employed to assess coastal vulnerability at a regional scale (Sallenger, 2000; Stockdon et al., 2007; Leaman et al., 2021). While these simple models provide extremely fast estimations

of the expected impact, their output is qualitative, only describing the general scale of impact. Moreover, simple empirical or behavioural models are commonly employed for probabilistic dune erosion risk assessments to enable fast run-times (Ranasinghe et al., 2012; Li et al., 2014a; Antolínez et al., 2019). However, their accuracy and adaptability can be limited due to their simplifications. On the other hand, process-based models, like *XBeach* (Roelvink et al., 2009), have shown great skill in reproducing morphological changes at the dune system during storms (Lindemer et al., 2010; McCall et al., 2010;

Vousdoukas et al., 2011; de Winter et al., 2015; Passeri et al., 2018), but are computationally demanding. This can become particularly problematic for large spatial scales or when a large number of simulations is needed, hindering their application in early warning systems or probabilistic risk assessments. To this end, data-driven methods based on statistical approaches can provide a good alternative to achieve faster estimations. Since dune erosion observations (with large spatial and temporal coverage) are scarce, a common approach is to generate synthetic data of dune erosion based on plausible storm conditions

using process-based modelling (Poelhekke et al., 2016; Pearson et al., 2017; Santos et al., 2019). Among the most used data driven methods in coastal applications are Bayesian networks (Gutierrez et al., 2011; Poelhekke et al., 2016; Beuzen et al., 2017; Giardino et al., 2019; Sanuy and Jiménez, 2021), which use conditional probabilities between system parameters to study cause and effect. While they can provide useful information on system dependencies (Beuzen et al., 2019), their shortcoming lies in the mandatory discretization of the parameters and their tendency for overfitting when used as predictive

tools (Beuzen et al., 2018).

In the last decade, Artificial Neural Networks (ANNs) have been increasingly employed in coastal engineering and management applications, due to their effectiveness in modelling complex non-linear systems with great speed. ANNs have been employed for predictions of beach seasonal changes (Hashemi et al., 2010), longshore sediment transport (Kabiri-Samani



et al., 2011; Güner et al., 2013), sandbar characteristics (Kömürcü et al., 2013; López et al., 2017), storm surge (Kim et al.,
2015) and coastal overtopping (van Gent et al., 2007; Verhaeghe et al., 2008; Chondros et al., 2021). Of particular relevance
to this study, Santos et al. (2019) tested the use of ANNs, among other statistical models, to predict changes in dune geometry
during storms at Dauphin Island in U.S.A. Due to the manageable size of the region of interest (~14 km), they simulated dune
response with a 2D-*XBeach* model, and they developed an ANN for each of the 200 transects they studied using only storm
conditions as the ANN input parameters. However, since these ANNs were bound to the pre-storm morphology of the
individual transects, they cannot be used for different beach and dune morphologies.

Here, we present a meta-model based on ANNs for the prediction of dune erosion volume (DEV) during storms at any location
on the Dutch coast (~260 km of dune fronted coastline) (see Figure 1). The model was trained based on a synthetic dataset
created with *XBeach-1D* models, for a large number of cases with different coastal morphology and storm conditions
combinations. To ensure that the training data sufficiently capture the variability of the observed coastal morphology and
possible storm conditions, while keeping the computational costs for the creation of the synthetic dataset at a manageable level,
input reduction techniques and probabilistic analysis were used. First, using clustering techniques, a set of 100 representative
typological coastal profiles (TCPs) were chosen from ~1,400 available elevation transects at the Dutch coast based on local
morphological and hydrodynamic characteristics (Athanasiou et al., 2021). Then, a simulator of synthetic physically realistic
offshore storm events was developed using marginal distribution and copula fitting methods on the observed storm variables
along the Dutch coast. Using a dissimilarity analysis, 100 storm events were chosen per station which were used to force
*XBeach* models for each of the 100 TCPs. The pre-storm morphological and hydrodynamic profile characteristics of each TCP;
the offshore storm conditions; and the simulated DEV were used to create a training dataset with 10,000 cases. Using the
developed training dataset, a 2-phase ANN metamodel was created, which used the pre-storm profile characteristics and
offshore storm conditions as inputs and provides a DEV estimate as output. The two phases of the meta-model compose of 1)
a classifier, which estimates if there is dune erosion (DEV > 0) or not (DEV = 0), and 2) a regressor, which estimates DEV in
the case that the classifier predicts dune erosion. Different ANN configurations were tested using a benchmark dataset based
on *XBeach* simulations, and the predictions of the final ANN configuration were compared against observed DEV during three
historic storms.

Section 2 describes the methods used to create the representative training dataset and the development of the 2-phase ANN
meta-model. The optimum configurations of the ANNs and their error statistics are presented in Section 3. In Section 4, the
limitations of the presented methodology and potential applications are discussed. Finally, a summary of the main conclusions
of the work is presented in Section 5.


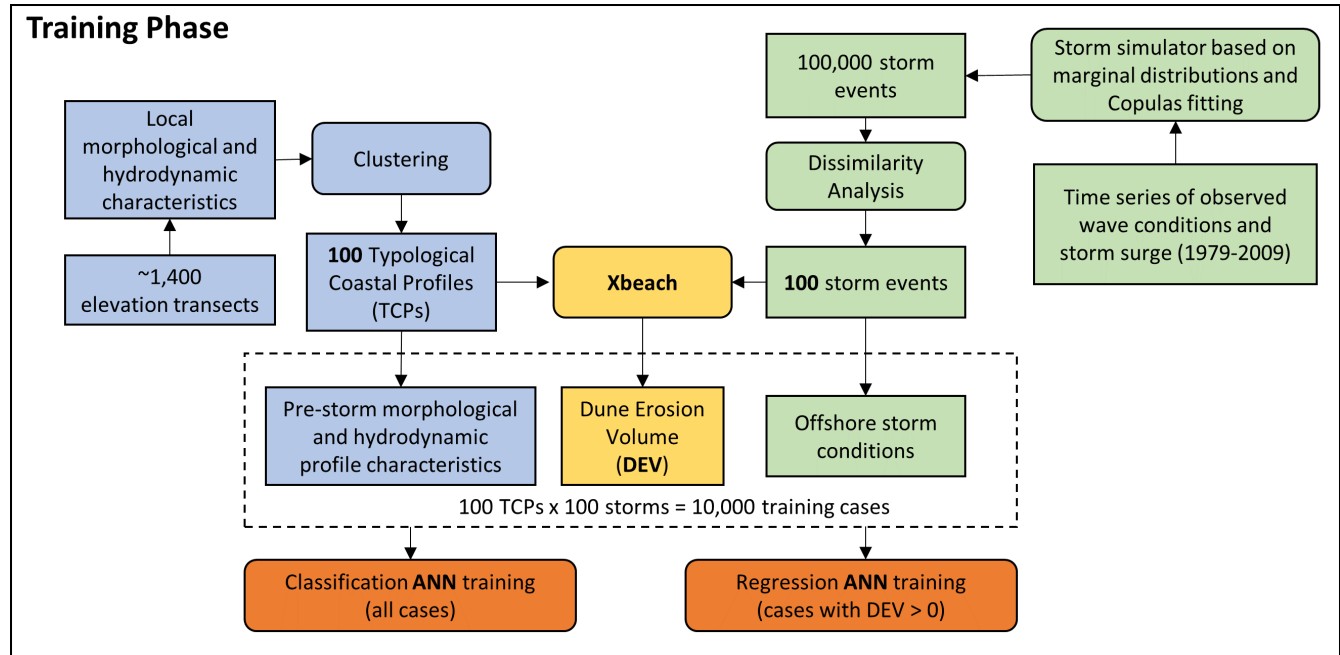

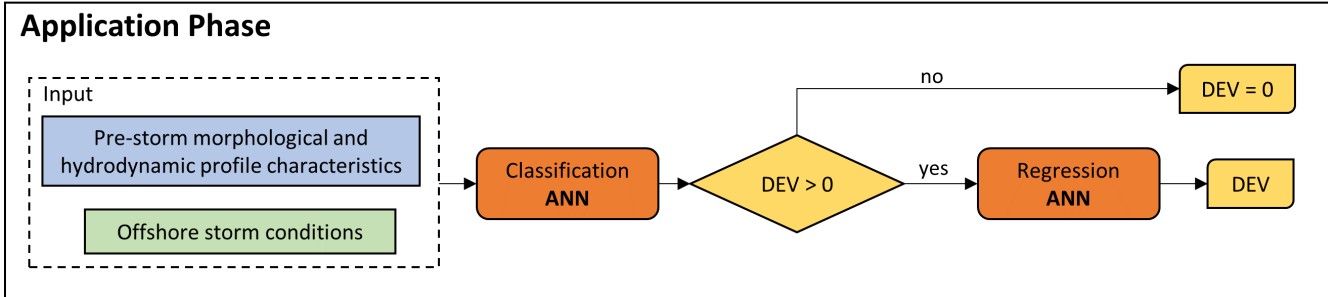

**Figure 1: Overview of the training and application of the proposed meta-model for dune erosion volume (DEV) predictions. Blue colours indicate steps associated with the profile characteristics, green colours indicate steps related to the offshore storm conditions, yellow colours the ones connected with dune response and orange colours the steps connected with artificial neural networks.**

## 2    Methods

### 2.1    Case study and data availability

The Dutch coast, located at the North Sea, has a total length of 432 km, including 1) the Delta region in the South, composed of islands and estuaries, 2) the Holland coast in the centre, with long stretches of sandy beaches and dunes and 3) the Wadden islands in the North, comprising barrier islands and tidal inlets (Figure 2). Almost 60% of the total coastline comprise beach and dune systems (Ruessink and Jeuken, 2002) , which form part of the coastal defences protecting the low-lying hinterland from flooding. Depth and elevation measurements are available on an annual basis (taking place between April and September)


at fixed transects along the Dutch coast through the JARKUS ("Jaarlijkse Kustmeting," Annual Coastal Measurement)
program.

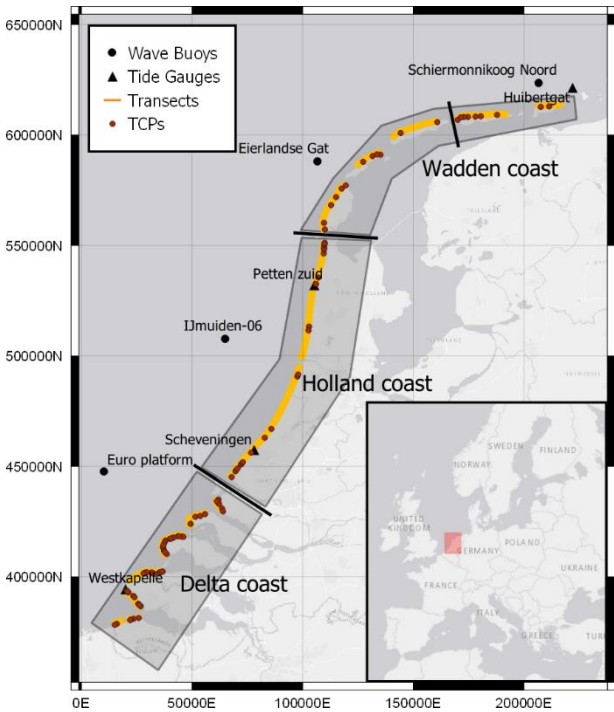

**Figure 2: Map of the Dutch coast indicating the location of the JARKUS transects (yellow lines), the offshore stations (wave buoys and tide gauges, black circles and triangles respectively), and the 100 typological coastal profiles (TCPs, red dots) used as training dataset in the present study. The thick black lines perpendicular to the coastline separate the different Dutch coastal areas, each one**
**associated to one offshore station. The map is projected on the "Amersfoort / RD new" coordinate system (EPSG: 28992) with the X and Y axis in meters. The basemap shown is obtained from ESRI (ESRI, 2011).**

Time series of wave conditions (significant wave height $H_s$, peak wave period $T_p$ and mean direction $Dir$) at 3-hourly intervals
are available between 1979 and 2009 from directional wave-rider buoys, at four offshore locations at depths of about 25 m on
average (Figure 2). Additionally, water level time-series at 10 min interval are available from four tide-gauges along the coast
(Figure 2), for which a tidal analysis was performed in Athanasiou et al. (2021) and the local tide and storm surge level ($SSL$,
difference between water level and tidal level) were extracted. The wave and water level time series are available from
Rijkswaterstaat, an agency of the Ministry of Infrastructure and Water Management of the Netherlands.

In Athanasiou et al. (2021) 1,430 of the JARKUS transects, that represent sandy beach and dune systems, were chosen and the
elevation profiles were extracted for the year 2019. In that study, various morphological and hydrodynamic parameters were
calculated per profile and episodic dune erosion volume was simulated per profile with the process-based model *XBeach* for
10 representative storms.





## 2.2 Typological coastal profiles

A 2-phase clustering approach similar to Athanasiou et al. (2021) was applied to group the available profiles into 100 clusters based on their similarities, each described by its representative profile (centroid) called a typological coastal profile (TCP) (red dots in Figure 2). We excluded some of the 1,430 profiles from Athanasiou et al. (2021), which were found to be at highly dynamic areas (e.g., transect at the tips of the Wadden islands), leaving 1,368 transect for our study. The 1,368 profiles were first clustered to 300 initial groups, leading to 300 initial TCPs, based on a set of 10 morphological and hydrodynamic parameters that characterize each profile (Table 1). Then a second clustering was performed on the 300 initial TCPs, where their simulated dune response to ten different storms was taken into account, to group them in 100 clusters and obtain the final TCPs. For detailed information on the exact methodology we refer to Athanasiou et al. (2021). Most of the resulting TCPs were located at the Delta coast (~55%), indicating that the similarity of the profiles there is lower, in comparison to the Holland (~25%) and Wadden (~20%) coasts.

**Table 1: Morphological and hydrodynamic parameters used to characterize each profile in the clustering procedure along with their individual weights. A brief description of each parameter is provided. For more detailed information please see Athanasiou et al. (2021).**

| Parameter | Unit | Description | Weight |
|---|---|---|---|
| Beach volume | m³/m | Subaerial sand volume, per alongshore running meter, seaward and above the MSL point. | 1.00 |
| Beach width | m | Cross-shore distance between the MSL and dune toe locations. | 0.95 |
| Beach slope | - | Linear slope between the MSL and dune toe points. | 0.95 |
| Nearshore slope | - | Linear slope between the depth of closure and the MSL points. | 0.70 |
| $a_{extreme, mean}$ | ° | Average angle of incidence at the depth of closure during historic extreme events | 0.70 |
| MHW | m | Mean high water level | 0.65 |
| $SSL_{RP100}$ | m | Storm surge level with a return period of 100 years | 0.35 |
| $T_{p, RP100}$ | s | Nearshore peak wave period with a return period of 100 years | 0.30 |
| $P_{x, extreme, mean}$ | KWh/m | Average nearshore wave energy flux toward the coast during historic extreme events | 0.25 |
| Max dune volume | m³/m | Subaerial sand volume, per alongshore running meter, seaward of the max dune crest and above the dune toe elevation. | 0.20 |

## 2.3 Selection of storm events

To ensure that our data-driven model will have predictive capacity for storms that might not have been directly observed in the offshore wave and water level records, synthetic storms, that are physically realistic according to the historic offshore climate, should be included in the training process. To accomplish this, first, marginal distributions were fitted to the storm parameters of observed extreme events and then, a copula-based approach was used to simulate a much larger number of synthetic events, while preserving the interdependencies between the different storm parameters. Here, extreme storm events





are defined as instances when $H_s$ and $SSL$, both exceed a threshold, making it more likely for dune erosion to occur. To capture the spatial variability of the hydrodynamic environment along the Dutch coast, we divided the coastal zone in four areas, each covered by the closest of the four wave stations (along with their closest tide gauges) as shown in Figure 2.

Using an approach similar to Wahl et al. (2016), we identified the $H_s$ and $SSL$ thresholds per area implicitly(2016) by identifying the $H_s$ and $SSL$ conditions when the total water level ($TWL$) exceeded an elevation threshold defined by the average dune toe elevation. We estimated the historic $TWL$ per area by adding the tidal level, the $SSL$ and the wave runup $R_{2\%}$ components. $R_{2\%}$ is estimated from offshore wave conditions and the average beach slope using the empirical formulation of Stockdon et al. (2006). Then the instances when the $TWL$ exceeded a critical threshold, which here was defined as the $10^{th}$

percentile of dune toe elevation in each of the four areas (to ensure that dune erosion occurs at some profiles), were identified and the annually averaged values of $H_s$ and $SSL$ during these instances were extracted (Wahl et al., 2016). The lowest values of $H_s$ and $SSL$ per area were used as the thresholds that defined the historic storm events per area, which ranged between 3.2-3.6 m for $H_s$ and 0.7-0.8 m for $SSL$. The approach we used to identify the historic extreme storms per area can be summarised as follows: (1) find the $H_s$ exceedances of the local $H_s$ threshold in the $H_s$ historic time-series; (2) calculate the duration ($D$) of

the event as the time period when $H_s$ remained above the threshold, while events with a time separation of less than 24 h were merged as one event assuming they are associated with the same storm (Wahl et al., 2016; Athanasiou et al., 2021); (3) extract the maximum $H_s$ during the event, and the concurrent $T_p$ and $Dir$; (4) extract the maximum $SSL$ during the event; (5) exclude events with a maximum $SSL$ less than the local $SSL$ threshold.

As mentioned before, the first step towards simulating plausible storm events is to fit marginal distributions to the extreme

storm parameters ($H_s$, $T_p$, $SSL$, $D$). Since in the dune erosion modelling approach we are using here, we assume that the wave direction is shore-normal (see section 2.4), the $Dir$ storm parameter is not taken into account in the event simulation process (except the step in the event selection process described previously that ensures that only storms directed to the coasts are taken into account). For the remaining storm parameters, we followed an approach similar to Li et al. (2014b), fitting a Generalized Pareto (GP) distribution to the storm parameters above a specific threshold. This threshold was identified by an

iterative process, where we tested different values of thresholds, in order to minimize the root mean square error (RMSE) of the fitted values against the observed values that had a return period larger than 1 year (ensuring that the fit is best for the extreme case i.e., tail of the distribution, for which we are mostly interested), following Li et al. (2014b). The shape and scale parameters of the GP distribution were obtained using a maximum likelihood estimation, and the goodness of fit was verified using the Kolmogorov–Smirnov test (Massey, 1951) considering a 5% significance level. Ultimately, the marginal distribution

of each storm parameter was described by its fitted GP distribution for the values above the threshold, and by its empirical distribution for the value below the threshold (Li et al., 2014b). This approach was applied to all the four storm parameters $H_s$, $T_p$, $SSL$, $D$ and was repeated for each of the four areas of the Dutch coast.

The simulated extreme events should preserve the observed dependency structure between the storm parameters, and to do so we followed a copula-based approach. Li et al. (2014b) compared four different methods to construct the dependency structure

of the offshore storm variables at the IJmuiden-06 station (see Figure 2), identifying the Gaussian copula as the best approach





due to both its performance and simplicity. The Gaussian copula, which belongs to the elliptical copulas family, is essentially a distribution over the unit-hypercube $[0, 1]^d$, and can be easily generalized to a higher number of dimensions. We first transformed the observations using their ranks normalized by a factor of $1/(n+1)$, where n is the number of observed events, and we then fit a Gaussian copula to the transformed observations, using the linear correlation coefficient between the variates.

Then using a Monte-Carlo scheme we generate 100,000 quadruplets in the unit-hypercube, which we then transform to their real units using the inverse cumulative distribution functions of the respective marginal distributions. To test if our simulator was able to capture the dependency structure between the storm parameters, we compared various dependency statistics (Pearson correlation ρ, Kendall's rank correlation τ, non-parametric tail dependence TDC) (Wahl et al., 2016) derived for the simulated and observed events for all the different combinations of storm-parameters. The relative differences between the

observed and simulated dependency statistics were smaller than 5% (Figure 3), verifying that the simulator was able to capture the dependency structure.

Finally, since we cannot simulate all 100,000 events with our process-based model (section 2.4) due to computational constraints, we used the maximum dissimilarity algorithm (MDA) (Camus et al., 2011; Athanasiou et al., 2021) to sample a representative subset of 100 events (Figure 3). We preferred to use the MDA relative to other input reduction approaches (e.g.,

K-Means), to ensure that the chosen events were sampling the extreme cases well enough (Santos et al., 2019). Another useful attribute of the MDA is that the most dissimilar cases are picked iteratively, based on their similarity with the rest of the cases. This means that e.g., the 50 first events from 100 events picked with the MDA will be the same as the 50 events picked with the MDA if a total of 50 samples were chosen. The number of events was chosen to ensure that the subset of events sampled the parameter space sufficiently, while keeping the simulation computational costs at a feasible level. This choice was similar

to the number of events used in similar studies for the creation of synthetic training cases (Poelhekke et al., 2016; Santos et al., 2019). The Copula-based Monte-Carlo simulation and the application of the MDA was applied to all four offshore stations (Figure 2), leading to 100 storm events per area.

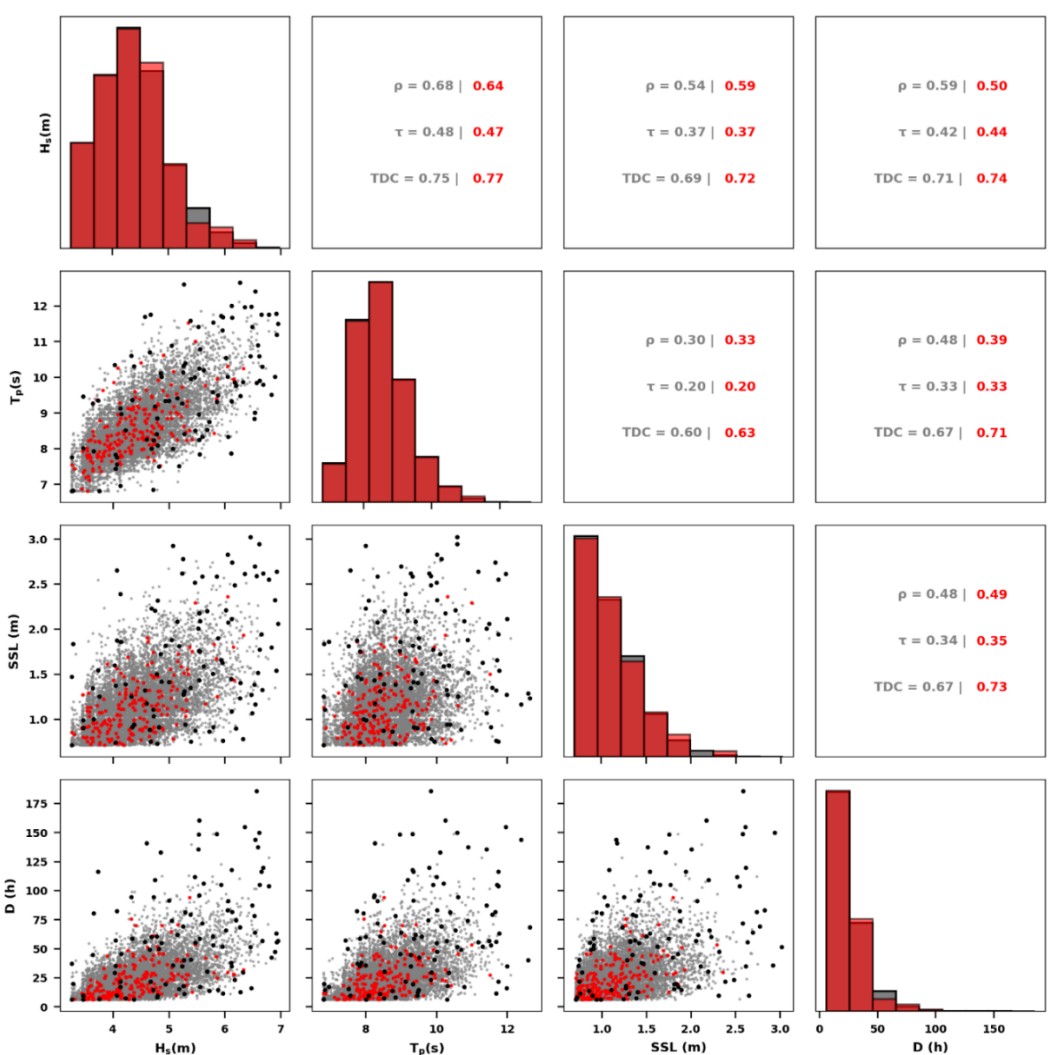

**Figure 3: Copula-based events simulator for the Euro platform station location (see Figure 2). Red, grey and black dots indicate observed, simulated and the 100 MDA selected events, respectively. The black dots are a subset of the grey dots (simulated events) that are selected with the MDA. Histograms of each storm parameter (*Hs, Tp, SSL* and *D*) for both observed and simulated events can be seen in the diagonal graphs. Below the diagonal, scatter-plots for each pair are plotted. Above the diagonal, 3 different dependency coefficients (ρ: Pearson correlation, τ: Kendall's rank correlation, TDC: non-parametric tail dependence) are shown for each pair for the observed and simulated events.**

## 2.4 Dune impact synthetic dataset

The process-based model *XBeach* (Roelvink et al., 2009) was used to simulate the dune response of the 100 TCPs (Section 2.2) during the 100 synthetic storm events (Section 2.3). *XBeach* is an open-source, process-based model which simulates short wave energy and long wave transformation, wave-driven currents, sediment transport and corresponding morphological changes at sandy coasts. In the present study, the *XBeach-v1.24.5867* version was used in a "surfbeat" mode, where the short-wave variations are resolved at the wave group scale. A one-dimensional (1D) cross-shore model was created for each TCP,





using the topo-bathymetry as extracted from the JARKUS dataset (extended to a 30 m depth assuming a slope of 0.01, to make sure that the offshore boundary was deep enough so that the waves at the boundary are in intermediate water). The cross-shore grid resolution changed from the offshore boundary to the dry beach, reaching a 1 m spacing at the dune area. An extensive calibration of the *XBeach* settings for use at the Dutch coast was performed in Deltares/Arcadis (2022), using both laboratory

experiments and field data (from Belgium, the Netherlands and Germany), resulting in a relative bias of -0.03 and a scatter index of 0.23 for the simulated dune erosion volumes. To reduce the computational time while ensuring minimal influence in computed morphological changes, we used a morphological acceleration factor (morfac) of 5, following Athanasiou et al. (2021).

For each of the 100 TCPs, the 100 synthetic storms derived for the respective offshore wave station (see Figure 2 and Section

2.3) were used to create the *XBeach* hydrodynamic boundary conditions. The time evolution of the boundary conditions was configured similar to Athanasiou et al. (2021), using a triangular approach for $H_s$ and $T_p$, and a normalized hydrograph for *SSL* (derived from the historic events of the respective station) which is rescaled using the individual storm's maximum *SSL*. The tidal signal was created for each TCP based on observed tidal records and the local *MHW*, assuming a high tide during the peak of the storm (centre of the hydrograph) (Figure 4a-c). The duration *D* was used to define the starting and ending time of

each simulation. Since the influence of wave obliqueness is not fully resolved in the 1D version of *XBeach*, here we assumed that the wave direction is shore-normal. Both assumptions, high tide during the peak of the storm and shore-normal waves can give a more conservative result with respect to dune erosion (see Discussion section). The synthetic wave time series were then used to construct JONSWAP spectra time series (gamma = 3.3 and wave directional spreading of 30°) that were imposed at the offshore boundary of the model, together with the superimposition of the *SSL* and tidal signal.

Here, the dune erosion volume (DEV) was chosen as the dune impact indicator (Giardino et al., 2014; Athanasiou et al., 2021) and was calculated for each simulation as the difference of the dune volume per running meter ($m^3/m$) above the highest water level during an event, between the pre- and post-storm elevation profiles (Figure 4d). For the cases were DEV was negative (resulting from storms with relatively small *SSL* and newly-formed dunes fronting the first dune row, leading to onshore sediment transport) or very small values ($DEV \leq 0.01$, values in this range are not expected to be representative

because of grid spacing limitations), the DEV was given a 0 value.

The synthetic training dataset comprised a total of 10,000 simulations (100 TCPs x 100 storms). Additionally, a synthetic benchmark dataset was created for calibration and validation purposes. This was created by simulating DEV with *XBeach* for 20 storms, selected using the MDA for each offshore wave station, but for a new Monte-Carlo simulation (to ensure that the events are different for the ones used in the training dataset) using 127 profiles along the Dutch coast (1 every 10 transects,

excluding the TCPs). The previous steps were followed to ensure that the benchmark dataset was independent from the training data but still accounted for the morphological and hydrodynamic variabilities observed at the Dutch coast. In this way, overfitting during model training was avoided, while the validation set was still representative enough. The benchmark dataset comprised a total of 2,540 simulations (127 profiles x 20 storms).


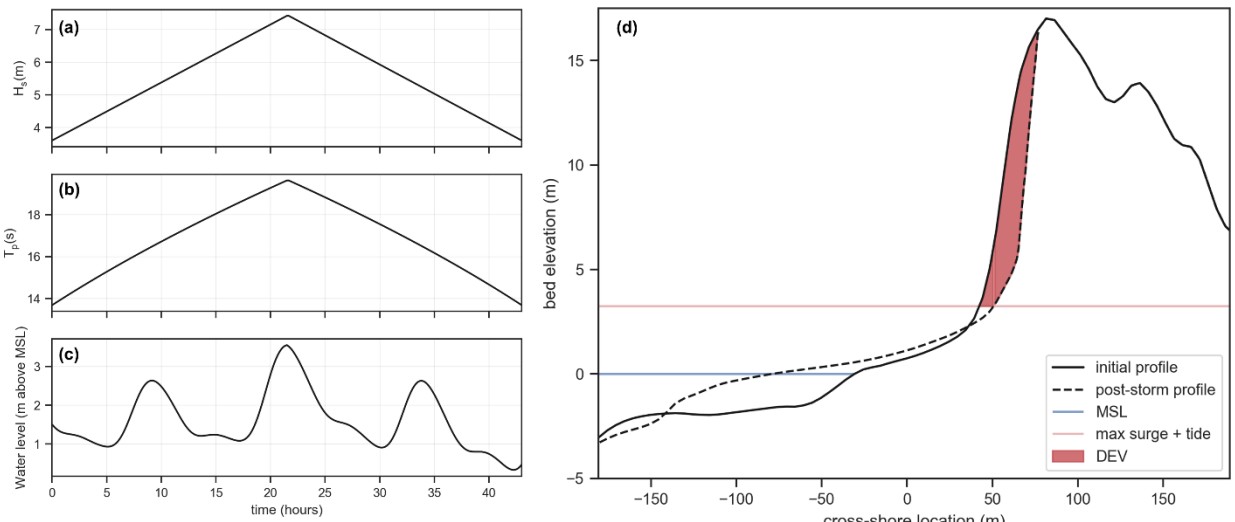

**Figure 4: Example of XBeach offshore boundary conditions and dune impact for one selected storm and TCP. (a) $H_s$ time series, (b) $T_p$ time series, and (c) water level time series at offshore boundary. (d) Post-storm simulated profile and calculation of dune erosion volume (DEV).**

## 2.5 Artificial Neural Network

Artificial Neural Networks (ANNs) are computational networks that are a subset of machine learning and more specifically deep learning algorithms, inspired by the neuron structure of the human brain. ANNs comprise a series of layers, each one containing fundamental units or nodes of a network. The number of layers of an ANN characterizes the depth and the number of nodes per layer the width of the network. More specifically, multi-layer perceptron feed forward ANNs include an input layer, one or more hidden layers, and an output layer; with the neurons of each layer being fully connected with the neurons of the previous one and the information passing in a single direction (i.e., from input to output). Thus, each neuron in the hidden and output layers translates the input values that are fed from the neurons of the previous layer to a single output, using a set of weights (one for each input neuron), a bias and an activation function. The output $y_j$ of the $j^{th}$ neuron of a layer with m neurons, is calculated as:

$$y_j = f\left(\left(\sum_{i=1}^{n} w_{ij} x_i\right) + b_j\right), with \ j = 1, 2, \ldots, m \tag{1}$$

where $n$ is the number of neurons in the previous layer, $w_{ij}$ is the weight of the connection of the $i^{th}$ neuron of the previous layer to the $j^{th}$ neuron of the current layer, $b_j$ the bias of the $j^{th}$ neuron and $f$ is an activation function.

Here, we built a 2-phase ANN for the estimation of DEV, composed of a classification ANN, which first predicts whether there is dune erosion or not, and then a regression ANN, which quantifies the DEV. With this 2-phase approach we ensured that overprediction for smaller DEVs was avoided and enabled the prediction of zero DEV (Verhaeghe et al., 2008). For the input layer a total of 14 parameters (neurons) were used (Figure 5), with 10 parameters describing the characteristics of the





pre-storm profile (which are the same parameters used for the clustering procedure in Section 2.2) and the 4 storm parameters as defined in Section 2.3. To avoid issues in the training of the ANN related to the different scales of the continuous input parameter space, the input data were first normalized to unity range. For both ANNs the initial weights and biases of the

neurons in the hidden and output layers were randomly assigned, and then the optimum weights and biases were calculated during an iterative training phase using the standard error-backpropagation with gradient descent method. More specifically, the adaptive moment estimation algorithm (Kingma and Ba, 2015) was used to calculate the optimum weights, by minimising a loss function between the ANN-estimated DEV values and the target values from the training dataset (the 10,000 cases described in Section 2.4).

The benchmark dataset (the 2,540 cases described in Section 2.4) was equally divided in a calibration and a validation dataset. The former is used for the tuning of the ANN's hyperparameters and comparing the different ANN configurations to choose the optimum one. The latter is used for assessing the actual accuracy and prediction skill of the selected ANN. To avoid overfitting of the model to the training dataset, the learning-phase of the ANNs was optimized by using the calibration dataset, to stop training when the losses (differences between predictions and targets) for the validation dataset stop decreasing.

Additionally, the hyperparameters of the ANNs, like the batch size (number of samples to work through before updating the internal model parameters), the learning rate (how much to change the model in response to the estimated loss each time the model weights are updated), and the hidden layer activation functions were chosen using a grid-search approach identifying the combination minimizing the calibration dataset losses. The division of the benchmark dataset to a calibration and validation dataset was performed 10 times with different randomization seeds and the final mean error statistics were used, to ensure that

any bias of the individual divisions was minimized. This meant that 10 different ANNs were produced (with the same architecture but different weights and biases), which will give an ensemble of DEV predictions, that can work as an uncertainty range.

For the neurons in the hidden layer(s) of both the classifying and regression ANNs, the rectified linear (ReL) activation function (Nair and Hinton, 2010) was used:


$$f(x) = \max\{0, x\} \tag{2}$$

For the classification ANN, all the cases in the training and benchmark datasets were used. The simulated DEV of all the training and benchmark cases were translated to 0 or 1, representing respectively no erosion ($DEV = 0$) or erosion ($DEV >$

$0$). For the neurons in the output layer the sigmoid activation function was used:

$$f(x) = \frac{1}{1 + e^{-x}} \tag{3}$$



With this function, the output neuron will always give a value in the range [0, 1] which indicates the probability of having

dune erosion and is used to classify the output to "erosion" if it is > 0.5 or "no erosion" if it is ≤ 0.5. Since the problem was a binary-classification one, the loss function used to optimize the classification ANN was the binary cross-entropy (BCE) defined as:

$$BCE = -\frac{1}{N}\sum_{i=1}^{N} y_i \cdot \log\big(p(y_i)\big) + (1 - y_i) \cdot \log\big(1 - p(y_i)\big) \tag{4}$$


Where N is the number of samples in the training batch, $y_i$ is the actual (observed) class, which is 0 or 1 for no erosion or erosion respectively, and $p(y_i)$ is the probability of the erosion class as calculated by the ANN.

For the regression ANN only the cases where $DEV > 0$ were used, which comprised 79% and 77% of the total cases in the training and benchmark cases, respectively. The DEV values were first transformed to $\log(DEV)$ to ensure that during the

calculation of losses, similar importance was given to minor and larger DEV cases (van Gent et al., 2007). For the output layer a linear activation function was used, which is defined as $f(x) = x$, and the final DEV estimation was acquired by using the exponential function on the output neuron to transform them back to real units. Since this problem was a regression one and the natural logarithm of DEV was used as the output, the loss function used in the regression ANN was the mean absolute error (MAE) defined as:


$$MAE = \frac{1}{N}\sum_{i=1}^{N}\big|\log\big(DEV_{pred}\big) - \log\big(DEV_{sim}\big)\big| \tag{5}$$

where N is the number of samples in the training batch, $DEV_{sim}$ is the dune erosion volume as simulated by *XBeach* and $DEV_{pred}$ is the dune erosion volume as calculated by the ANN.

Different architectures of the hidden layers (number of hidden layers and neurons per layer) were tested for both ANNs to find the optimum ones with respect to the calibration dataset. The accuracy metric (i.e., the percentage of correct class predictions) was used to find the optimum architecture of the classification ANN. For the regression ANN, the skill score (Murphy, 1988) was used to choose the optimum architecture, which is defined as:

$$skill = 1 - \frac{RMSE^2}{\sigma^2_{DEV_{sim}}} \tag{6}$$

$$RMSE = \sqrt{\frac{\sum_{i=1}^{N}\big(DEV_{pred} - DEV_{sim}\big)^2}{N}} \tag{7}$$


where $\sigma_{DEV_{sim}}$ is the standard deviation of the *XBeach*-simulated DEV in the calibration dataset and N is the number of cases

in the calibration dataset. A skill score of 1 indicates a perfect prediction. The bias index (*BI*), *RMSE* and the modified index

of Mielke (λ) (Duveiller et al., 2016; Santos et al., 2019) were also computed as complementary error statistics.

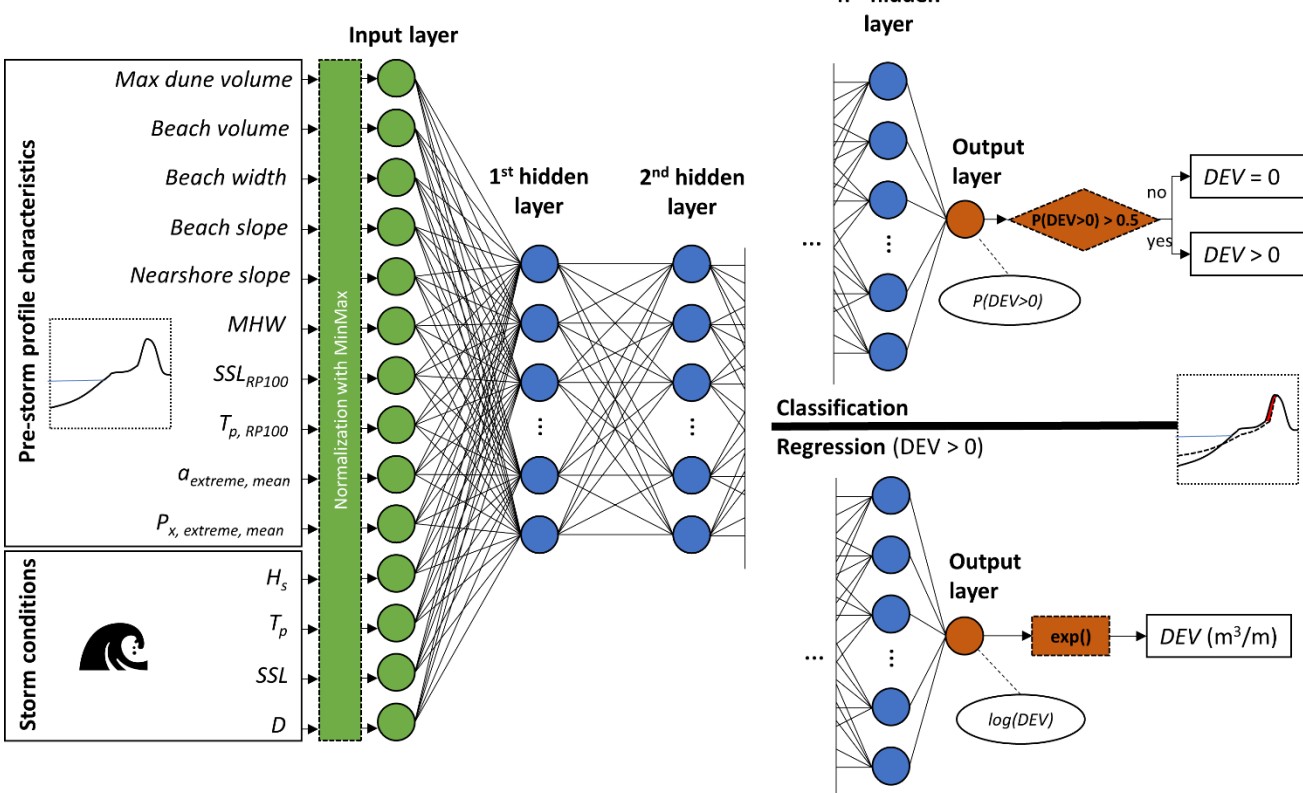

**Figure 5: Configuration of the artificial neural network (ANN) for either DEV classification or regression. The first layer (input layer) comprises 14 (10 profile and 4 forcing) parameters, which are normalized. Then, n hidden layers with variable numbers of**
**nodes are defined iteratively, while connections to the nodes of the previous layer are established using weights, biases and an activation function. The last layer is the output layer for which there can be two configurations: 1) Classification (upper right), where the output layer comprises 1 node presenting the probability of DEV > 0, which provides an estimate of dune erosion occurrence or not, and 2) Regression (lower right), where the output layer comprises 1 node, which is the log(DEV) that is translated back to its real units using an exponential function.**

## 3    Results

### 3.1    Performance of Classification ANN

Different depths (number of hidden layers) and widths (number of neurons per layer) of the ANN were tested to find the

optimum architecture of the classifier. Architectures with one, two or three hidden layers were tested, with similar number of

neurons ranging between 2 and 128 (with a quadratic step increase between the tests). The mean accuracy of the predictions

for the calibration cases (1,270 cases) increased for wider networks, starting from an average accuracy of ~85% for a small
number of neurons per layer and reaching up to ~95% for wider networks (Figure 6). The difference between networks with
one and three hidden layers was negligible, while for networks with more than 16 neurons per hidden layer the relative increase
in accuracy was again negligible. This indicated that the accurate binary classification for the prediction of whether dune
erosion occurs or not does not necessarily require a complex ANN. Therefore, and since the differences between the different

architectures beyond a threshold of 16 neurons per layer were in the same order of magnitude as the expected differences of
rerunning the algorithm (due to the inherit stochasticity of the training process and the weight initialization), here we opted to
use a relatively simple ANN architecture with 2 hidden layers of 32 neurons each.

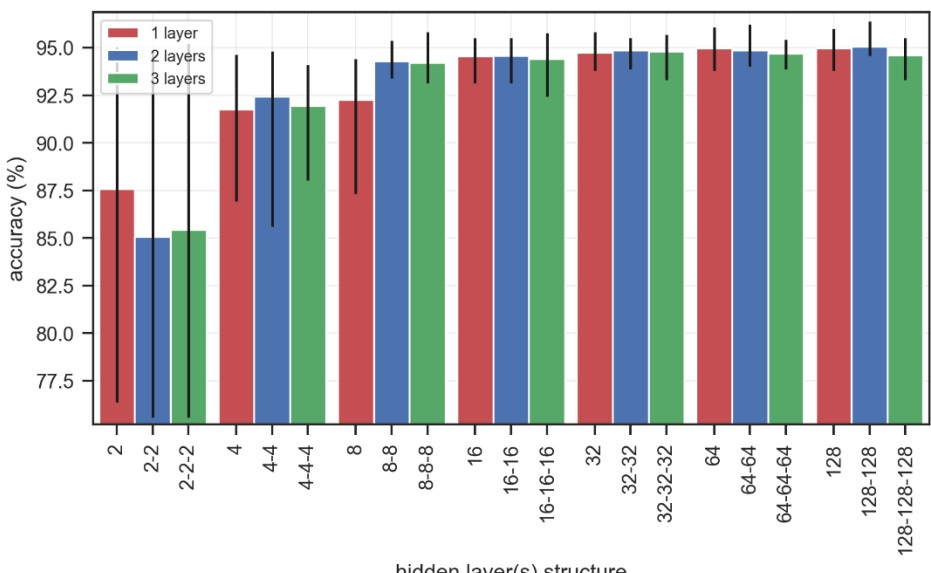

**Figure 6: Accuracy of classification prediction for the calibration cases using different architectures of the classification ANN with**
**different number of hidden layers and neurons per layer. In the x-axis the architecture of each ANN is represented by the number**
**of neurons per layer, with layers separated by "-". The bars indicate the mean accuracy values between the 10 randomisations of**
**the benchmark dataset, and the lines show the min and max values.**

Using the 1,270 validation cases the accuracy of the selected ANN was further assessed. The classification ANN predicted the
correct class ("erosion" or "no erosion") with an average accuracy of ~94%. The confusion matrix shows that the model had

an average accuracy of ~96% in correctly identifying cases with erosion, while it accurately found ~85% of the cases that there
is no erosion (Figure 7). The quite high accuracy in identifying erosion cases provided confidence in the use of this
classification ANN as a first filtering step in the proposed meta-model, while the "false positive" cases (~15%) are handled in
the next phase with the regression ANN, where they can still have a small DEV value. Since the actual output of the
classification ANN is the probability of erosion, a prediction confidence can be calculated as the probability of the class with

the highest probability. When analysing the statistics of the prediction confidences (Figure 7), it can be seen that for accurate
predictions (true-positive or true-negative) the prediction confidence was higher ($25^{th}$-$75^{th}$ percentiles: 99.7-100%) than that


for false-negatives (63.7-94.1%) and false-positive (67.1-97.2%), indicating that the prediction confidence as given by the ANN is a good measure of the confidence level of the predictions.

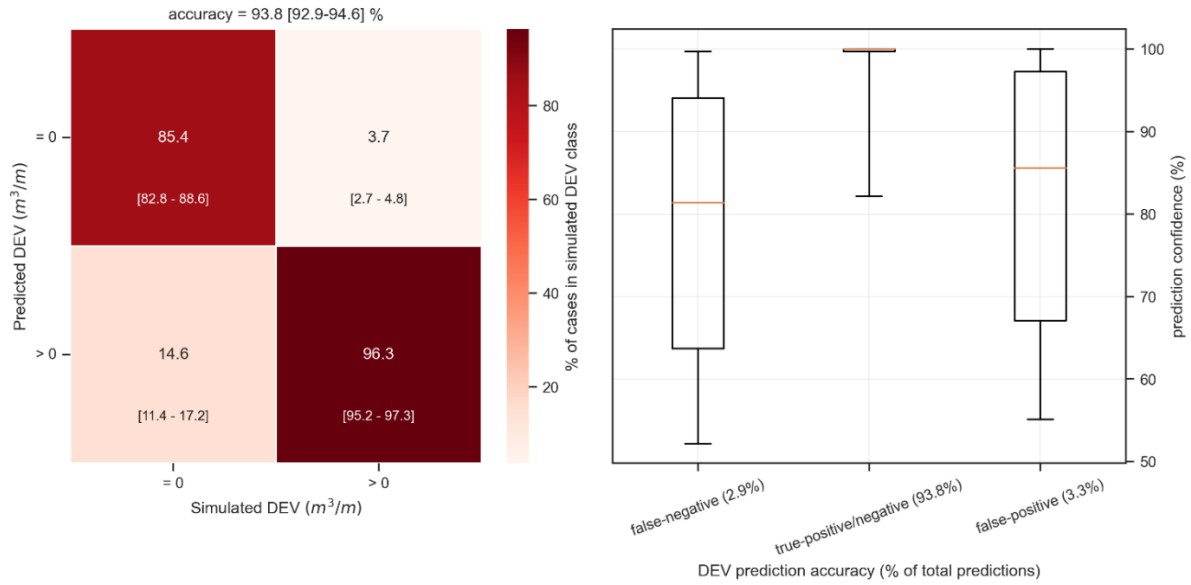

**Figure 7: Prediction accuracy of the predicted DEV class using the selected classification ANN configuration (2 hidden layers of 32 neurons each) against the XBeach simulated classes for the validation cases. (Left) Confusion matrix showing the percentage of each class prediction per observed class (each column sums to 100%), with the diagonal indicating a correct prediction. Numbers in the centre of each cell show the mean accuracy over the 10 randomizations of the benchmark cases, while brackets in the lower part give the min-max values. (Right) Boxplots of the prediction confidence (as given by the ANN) grouped by the prediction's accuracy,**
**defined here as the difference between the predicted DEV class and the observed DEV class (orange line: median, box: 25th-75th percentile range, lines: 5th-95th percentile range).**

### 3.2 Performance of Regression ANN

The same architectures as for the classification ANN were tested for the regression ANN. The mean skill score of the predictions for the calibration cases (~980 cases with DEV>0) strongly increased for wider networks, while the influence of
the network's depth was more evident for less wide networks (Figure 8). The consistency of the skill score between the calibration randomizations (shown by the line around the bars in Figure 8) increased with wider and deeper networks as well. This indicated that for a skilful prediction of DEV a relatively higher number of neurons (and thus complexity) was needed. Ultimately, a regression ANN with 3 hidden layers and 32 neurons per layer was selected as the best performing one.


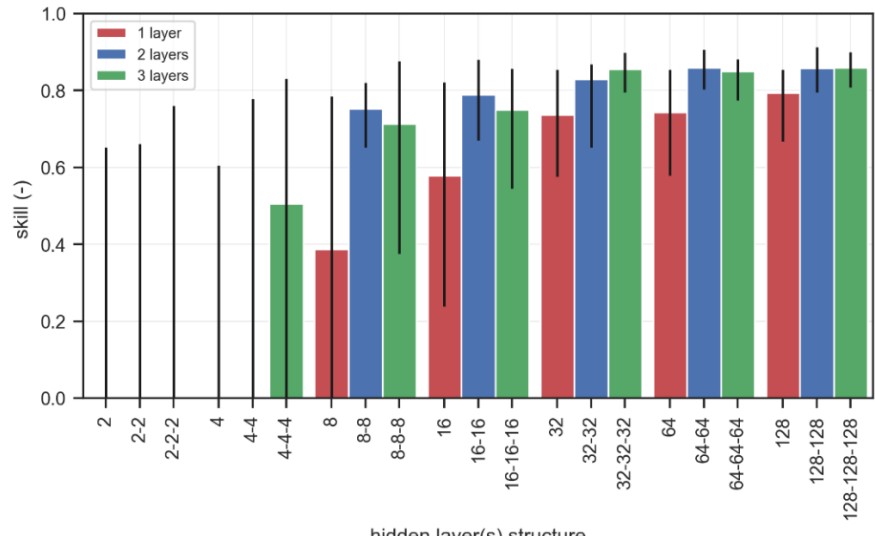

Figure 8: Skill of the DEV prediction for the calibration cases using different configuration of the regression ANN with different number of hidden layers and nodes per layer. In the x-axis the architecture of each ANN is represented by the number of neurons per layer, with layers separated by "-". The bars indicate the mean values between the 10 randomisations of the benchmark dataset, and the lines show the min and max values. The lower limit of the y-axis has been set to 0, to avoid visualizations issues when skill was negative for some configurations with low number of layers/nodes.

Using the validation cases (~980 cases) the error statistics of the DEV predictions with the selected ANN configuration were further assessed (Figure 9). The ANN-based DEV predictions showed a good agreement with the *XBeach* simulated DEVs, with a skill score between 0.71-0.89 (among the 10 randomizations) with an average of 0.82, while the RMSE and the bias had an average value of 19.4 and 0.23 $m^3/m$ respectively.



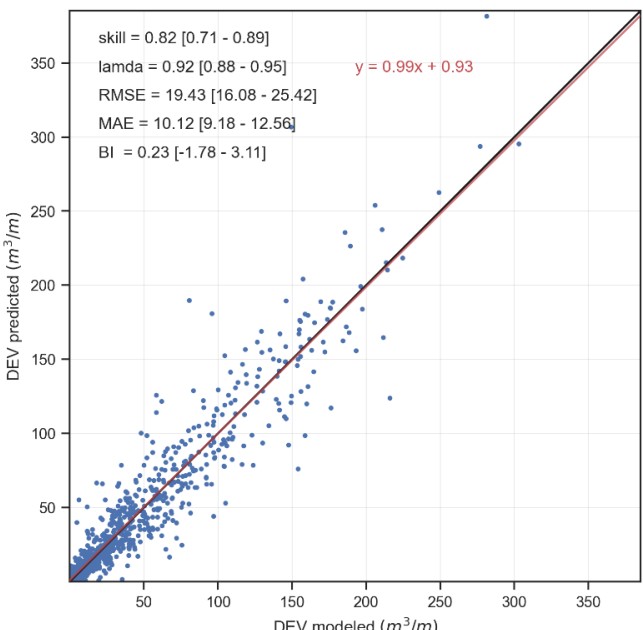

**Figure 9: Scatter plot between the XBeach simulated (x-axis) and ANN predicted (y-axis) DEV, for the validation cases with DEV > 0 m³/m, using the selected regression ANN configuration (3 hidden layers of 32 neurons each). The scatter plot shows the validation cases for one of the randomizations while statistics in the upper left show the mean over the 10 randomizations of the benchmark cases, with brackets giving the min-max values. The black line indicates the 1:1 line, while the red line is the linear regression fit (equation given in red).**

## 3.3 Comparison with observations

In the JARKUS program, elevation profiles are measured along the Dutch coastline every year around April as described in Section 2.1. However, these profiles cannot directly be used as pre- and post- storm profile observations to find dune erosion, since several different events and dune recovery occurs between the two measurements.

In this section, we used available pre- and post-storm observations for three historic storms (1953, 1979 and 2019) to validate the prediction capacity of the proposed meta-model against observed DEV. These three storms have been used in the validation of the *XBeach* model as well (Arcadis/Deltares, 2022). For these storms we extracted the storm conditions using the same methodology as used in the event definition in Section 2.3. The storm conditions for three different storms along with a description of the available elevation data is given in Table 2. The input for the ANN-based meta-model was obtained by calculating the morphological indicators of the pre-storm profiles, while the hydrodynamic indicators were obtained from the corresponding transects of interest as calculated in Athanasiou et al. (2021) (see Table 1).

**Table 2: Cases used for the validation of the meta-model predictions.**

| Storm | SSL (m) | Hs (m) | Tp (s) | D (h) | Number of transects | Remarks |
|---|---|---|---|---|---|---|
| | | | | | | |





| February 1953 | 3.0 | 7.3 | 14.1 | 37 | 1 | A typical schematized profile of the Holland coast is used as the pre-storm profile, while for the observed dune erosion, reported values of 90±26 m³/m are used (Van Thiel de Vries, 2009). |
|---|---|---|---|---|---|---|
| January 1976 | 2.2 | 6.1 | 10.8 | 35 | 30 | Pre- and post-storm elevation data at the northern part of the Holland coast. These data account only for the profile part up to the new dune toe. Final pre- and post- storm profiles are created by combining the previously described data with the JARKUS elevations of 1975 for the none measured points (Arcadis/Deltares, 2022). |
| January 2019 | 1.3 | 5.2 | 11.1 | 27 | 7 | Complete pre- and post-storm beach/dune profiles near Egmond aan Zee available from Ruessink et al. (2019). |

A scatter plot of the observed versus the ANN-predicted DEV for the different storms and transects (Figure 10) reveals that the variability in DEV values between the three storms was modelled well. For the moderate intensity storm of 2019, the

variability between the different measured transects was captured accurately. However, for the case of the 1976 storm there was some under-estimation for the transects with observed DEV $> 40 \ m^3/m$ and some over-estimation for the transects with DEV $< 40 \ m^3/m$. This might be an artefact, of the way the pre- and post-storm profiles were created, in that the 1976 storm observations only included the part up to the dune toe. For that reason, part of this discrepancies can be attributed to the quality of the measured data. For the disastrous event of 1953, the mean DEV prediction was in the range of the reported DEV values.

Using these 38 cases, error statistics were computed for the mean DEV prediction of the ANN-based meta-model. The DEV predictions had a skill score of 0.84 and RMSE of 9.15 $m^3/m$ against the 38 observed DEV. The error statistics against the observed cases (Figure 10) are comparable to the ones derived against the benchmark cases (Figure 9), with the exception of the RMSE, which is almost half the size. This is due to the relatively smaller scale of observed DEV in comparison to the simulated DEV in the benchmark dataset.



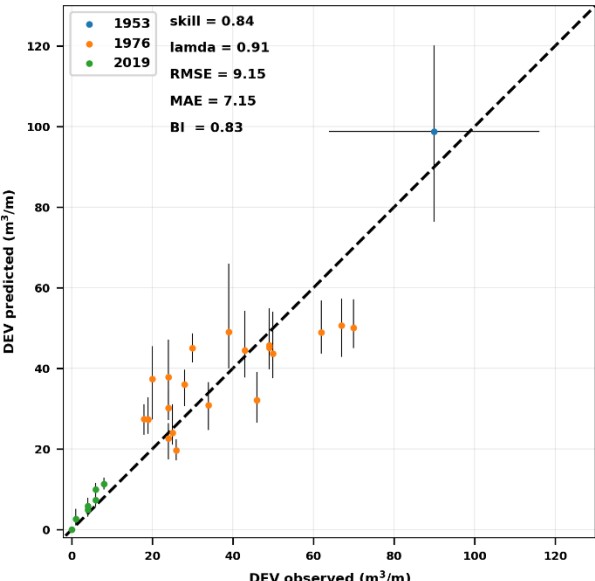

**Figure 10: Scatter plot between the observed (x-axis) and ANN-predicted (y-axis) DEV, for three historic storms with variable number of transects per storm. The dots indicate the average predictions of the ANN, while the vertical lines show the min and max values as given by the ensemble of the ANN output. For the storm of 1953 the horizontal line gives the range of the values reported.**

## 4    Discussion

### 4.1    Insights

During the creation of the ANN-based meta-model, a specific set of profile and storm parameters was used as input features for the DEV predictions (Figure 5). The 10 profiles parameters used herein were the same ones used in the clustering procedure as presented in Athanasiou et al. (2021), which were found to drive DEV variability during storms at the Dutch coast. During the clustering procedure (Section 2.2), specific weights were assigned to each of these 10 parameters according to their

importance (Table 1). These weights, however, were not used in the ANN creation, because a weighting scheme is implicitly taking place during the training of the ANN, when weights are assigned to each incoming connections of each individual neuron. Since there are numerous connections and different layers, it is not straightforward to get individual weights for each of the input parameters of the trained ANN as a measure of input importance. Still, there are techniques that allow for assessing the individual impact of inputs in an ANN (Wang et al., 2000).

Here we employed a permutation importance approach (Breiman, 2001; Logue et al., 2019) to assess the individual importance of the input parameters in predicting DEV using the ANN framework. To this end, the trained regression ANN was used to predict DEV for the validation cases, after each feature (i.e., input parameter) was first randomly permuted one at a time. Then, the MAE was calculated for each permutation and compared with the benchmark MAE (using the validation cases without permutations) (Figure 11). The underlying idea is that if an input feature is important, then shuffling its values will lead to

larger errors in the predictions. The pre-storm beach volume was found to be the most important input in predicting DEV. The
other beach characteristics (width and slope) were important as well, highlighting that the beach fronting the dune act as a first buffer protecting the dune from erosion. The storm's $SSL$, $T_p$ and $D$ were found to be quite important inputs as well (in that order). These results agree with Beuzen et al. (2019), who found beach width and the exceedance of wave runup above the dune toe as the most important drivers of observed dune erosion variability during a single storm along the southeast Australian

coastline. The Max dune volume was found to be the least important input for the DEV predictions, which agrees with Athanasiou et al. (2021). It should be noted though, that some of the parameters that are expected to be important, e.g., the storm's $H_s$, did not lead to a high MAE difference, something that could be connected with their correlation to other more important parameters (see Figure 3), that mask part of the individual importance during the permutations.

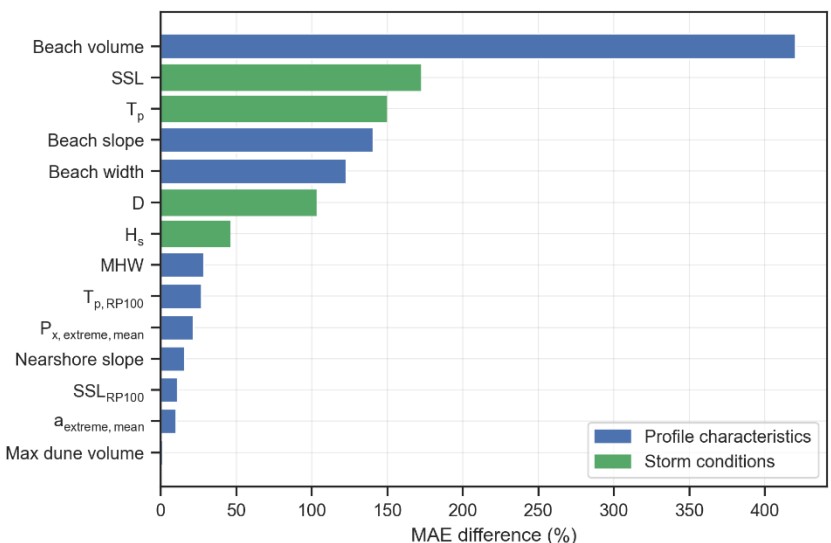

**Figure 11: Input parameter importance using a permutation importance approach. The bars indicate the mean MAE difference between the MAE when permutating a single input parameter and the MAE of the benchmark cases with no permutations.**

One of the most critical aspects when training machine learning models, like the ANN presented here, is the training dataset. The predictive capacity of the ANN is highly dependent on how well the training cases sample the parameter space, allowing for generalization of the model and avoiding overfitting. To ensure a representative (for the Dutch coast) training dataset, here

100 elevation profiles (TCPs) were chosen based on a clustering analysis of all the available transects at the Dutch coast (Athanasiou et al., 2021), while for the storms, a total of 100 storms was selected based on previous studies and available computational power. However, since for the selection of storm events, the MDA algorithm was used, which does not randomly pick events from the storm parameter space, but rather incrementally includes the most dissimilar events in the chosen subset (i.e., the order the events are picked is always the same); there is a chance that not all of the 100 events used here were needed

to capture the variability of the synthetic storms produced in Section 2.3. To test this, we trained the regression ANN with a different number of cases, representing the combination of the 100 TCPs with a different number of events as given by the MDA algorithm (5, 10, 20, 50 and 100 storm events) and validated the prediction skill of each ANN (Figure 12). The prediction





skill reached a plateau after 20-50 storm events were used. This means that with around 20-50 events, the MDA algorithm has already picked the most dissimilar events.

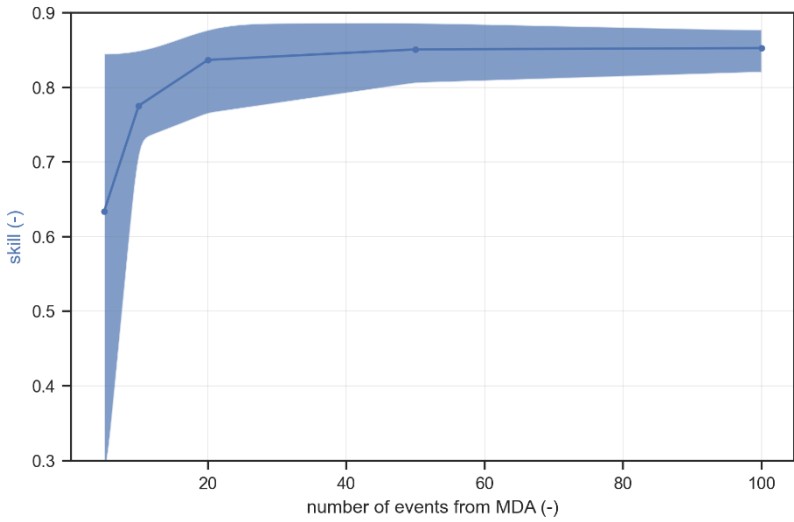


**Figure 12: Skill value for different number of training storm events used from the MDA algorithm (5, 10, 20, 50 and 100). The thick line shows the mean values while the patch indicated the min and max values between the 10 randomizations of the validation dataset. The patch extents have been smoothed using a Gaussian smoother.**

In our study, running 10,000 *XBeach* simulations for the training dataset, was feasible using parallel computing. However, as
shown in Figure 12, more than half of these simulations were actually not necessary to attain the predictive skill that was demonstrated in Section 3.2. To this end, this "spare" computational effort for creating a synthetic training dataset could be devoted to add complexity to the meta-model, by reducing some of the assumptions used in the presented framework. For example, 2D-*XBeach* models could be used instead of 1D, enabling the simulation of oblique waves during storms, by adding the wave direction as an extra input parameter in the ANN. Additionally, the assumptions with respect to the time evolution
of the water levels and wave conditions during a storm used here (see Section 2.4) could be replaced by expanding the simulations with different $H_s$ and $SSL$ hydrographs and different phasing between the peak of $SSL$ with the high-tide. The effects of the aforementioned assumptions are further discussed in the next subsection.

## 4.2    Limitations

Machine learning models based on supervised learning, like the presented ANN-based meta-model, are as good as the data
used to train them. Due to lack of a sufficiently large  number of post-storm dune erosion observations at the Dutch coast, the training dataset used here was a synthetic dataset created using a calibrated version of the process-based model *XBeach*, which has shown good skill in predicting dune erosion at the Dutch Coast (Arcadis/Deltares, 2022) and has been previously used in numerous studies showing good skill in reproducing dune erosion during storms (Lindemer et al., 2010; McCall et al., 2010; Vousdoukas et al., 2011; de Winter et al., 2015; Passeri et al., 2018). Still, due to the computational burden of creating the



synthetic cases, specific assumptions and simplifications needed to be followed, which can affect the predicting capacity of the presented meta-model.

One of the main simplifications was using a 1D model and assuming that waves are shore-normal. This was decided, due to the high computational cost of running ~10,000 2D-*XBeach* models, which can handle the incoming wave obliqueness. Nevertheless, alongshore non-uniformity and incoming wave angle can affect the dune erosion intensity due to generation of

alongshore currents and sediment transport gradients (De Winter et al., 2012). Additionally, since with the copula-based approach used here (Section 2.3), we could only simulate the main storm parameters ($H_s, T_p, SSL$ and $D$), the time-evolution of the wave characteristics and water-levels used in the *XBeach* boundary conditions needed to be schematized. Here we used a triangular hydrograph for the time evolution of the significant wave height and a representative normalized hydrograph based on historic events for the time evolution of storm surge, while it was assumed that the high tide always occurs at the same time

as the surge peak. While these choices were based on a previous study showing good agreement with observed parameters that can drive dune erosion (Athanasiou et al., 2021), these assumptions have been previously found to introduce extra uncertainties in the modelling of dune impacts (Duo et al., 2020). Some of these assumptions might explain some of the differences between the predicted and observed DEV for the historic cases (Figure 10).

The choice of the profile and storm parameters used as input in the ANN was based on previous studies looking into drivers

of dune erosion variability (Beuzen et al., 2019; Cohn et al., 2019; Athanasiou et al., 2021). However, there are other variables that can be of importance at other dune-fronted coastlines around the world. For example, sediment grain size effects were not taken into account in the present study, since the $D_{50}$ variability at the Dutch coast is not high enough and is not expected to drive strong differences in DEV variability (Athanasiou et al., 2021). Additionally, vegetation characteristics can play an important role in controlling dune erosion intensity during extreme events (Charbonneau et al., 2017) and were not studied

here. Moreover, since the training data of the ANN comprise a set of parameters that are defined in a quite specific manner (see Table 1), it is critical to follow the exact same definition when making predictions, since the model is data-driven rather than process-driven.

The width and height of the dunes along the Dutch coast results in the former being in the collision regime (Sallenger, 2000) during storms and overwash or inundation is extremely rare. The dune response indicator used in the present study was the

dune erosion volume, which is a good indicator to characterize the scale of erosion during extreme storms and provides useful information in long-term planning of nourishment strategies (Giardino et al., 2014). However, several flooding or erosion indicators can be of relevance for other applications or locations (Leaman et al., 2021), such dune retreat or dune crest lowering (Santos et al., 2019). The presented techniques could be expanded for these impact indicators.

### 4.3  Applications

The proposed meta-model could have various applications in coastal zone management due to its accuracy and efficiency in generating output. A prediction of DEV for all sandy transects of the Dutch coast, when a forecast of an incoming storm is available, takes a matter of seconds using this model. This highlights its potential use in an early warning system, compared



to the computational power needed when running *XBeach* for all transects, leading to $10^3$-$10^4$ times faster computations depending on the storm's duration. A hybrid approach could be followed as well, where the ANN meta-model could be used as a screening tool to identify hotspots of dune erosion, reducing the number of *XBeach* models runs needed to only the locations that are expected to have higher impacts.

Another potential application could be in dune erosion risk assessments, where the meta-model could be applied to quickly quantify DEV along the Dutch coast for all historic storms observed from the offshore stations. Then with extreme value analysis at each transect, local extreme values of DEV could be derived which can aid in assessing local coastal protection. Moreover, if the presented framework is expanded to predict dune retreat and include sea level rise, it could be used for sea level rise induced coastal retreat assessments as well, using the PCR tool (Ranasinghe et al., 2012). By efficiently picking the storm boundary conditions to model for the creation of the training data (Figure 12), extra computational power will be available to include sea level rise (SLR) scenarios. For example, with 100 TCPs, 20 synthetic storms per TCP and 5 SLR scenarios (e.g., 0.2, 0.4, 0.6, 0.8, 1), a total of 10,000 simulations will be needed. The SLR could directly be imposed in the offshore boundary of the new *XBeach* runs. Then a similar ANN-based meta-model could be trained with the extra input parameter of SLR and having dune retreat as output The PCR tool has been previously applied to one location at the Dutch coast (Li et al., 2014a) using an empirical relationship to estimate dune retreat during extremes. This probabilistic model requires a quite high number of simulations of coastal retreat, something that could be easily handled with the meta-model approach presented here.

The presented framework could be also expanded to enable the assessment of nourishment strategies. As mentioned in the previous section, the predictive capacity of an ANN is dependent on the training data. This means that elevation profiles with a sand nourishment might not look like any of the TCPs used in the training dataset. To address this, schematic profiles with various nourishments options could be created, then simulated with *XBeach* and finally added to the training cases. This will allow for more versatility in the predictive capacity and application range of the technique.

Finally, the ANN-based meta-model presented here is tailored to the Dutch coast and applying it anywhere else in the world would not be straight-forward (except where the elevation profiles and forcing look similar). Particularly, the training data used here to develop the meta-model were specific for the Dutch coastal setting with respect to coastal morphology and storm conditions. This means that the predictive capacity of the meta-model is dictated by the training data coverage of the input parameter space. However, the methodological framework (Figure 1) could be upscaled by creating a new training dataset, which would be based on *XBeach* simulations of a larger set of profiles that are representative of the global range of beach/dune morphologies and forcing conditions. When doing so, care should be taken to potentially introduce other critical input parameters, which can vary globally, like the sediment grain size. A meta-model like this could be used as a fast screening tool for dune erosion estimation at any study site with dunes around the globe, especially at data poor locations. However, availability of dune erosion data will remain a limiting factor for validating the model.





## 555   5    Conclusions

A meta-model based on artificial neural networks (ANN) and the process-based model *XBeach* has been developed to derive estimates of dune erosion volume (DEV) at the Dutch coast. The model can provide DEV predictions for the entire Dutch coast in a matter of seconds when compared to brute-force modelling with *XBeach*. The first step of this 2-phase model was to predict whether dune erosion occurs or not, using a classifying ANN which showed a mean accuracy of 94%, against a

benchmark dataset created with *XBeach* simulations. The second step comprised a regression ANN which estimated DEV with a mean prediction skill of 0.82 and RMSE of 19.4 $m^3/m$ against the benchmark data. Additionally, the predictive capacity was tested against some sparse observations of dune erosion for three historic storms at the Dutch coast, showing similarly good performance.

Input reduction techniques were used to ensure that the training data used for the ANNs included representative combinations

of elevation profiles and storm conditions. The input parameters of the meta-model included local morphological/hydrodynamic parameters and a set of storm-specific forcing conditions. Moreover, various ANN architectures were tested to find the optimum classification and regression ANNs. Using an input importance analysis, the local beach geometry, storm surge level, peak wave period and duration of the storm were found to be the most important inputs for more accurately predicting DEV. Potential applications of the presented model, include early warning systems and probabilistic risk

assessments of dune erosion, while the methodology could potentially be upscaled using more data from other beach/dune coasts around the world.

### Data availability

The resulting data can be obtained from the first author upon reasonable request.

### Author contributions

PA, AD, AG, MV, RR, and JAAA contributed to the design of the study. PA carried out the data processing, analysis, figure generation, and prepared the manuscript with contributions from all co-authors. All authors contributed to the article and approved the submitted version.

### Competing interests

The authors declare that they have no conflict of interest.



## Funding

This work has received funding from the EU Horizon 2020 Program for Research and Innovation, under grant agreement no 776613 (EUCP: "European Climate Prediction system;" https: //www.eucp-project.eu). AD was funded in part by the Deltares Strategic Research Programme "Natural Hazards", while AG from the Research Programme "Seas and Coastal Zones". RR is supported by the AXA Research fund and partly supported by the Deltares Research program "Seas and Coastal Zones".

## Acknowledgments

The Figures were created with Python 3.7.3 (https://www.python.org) and Matplotlib v3.1.2 (https://matplotlib.org/) libraries. For the creation and configuration of the artificial neural network, Keras (Chollet and others, 2015) and Tensorflow (Abadi et al., 2016) libraries were used.

## Disclaimer

The views expressed in the article do not reflect the views of ADB.

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
