# Peer review of "Estimating dune erosion at the regional scale using a meta-model based on Neural Networks"

_Natural Hazards and Earth System Sciences, 2022_

## Referee Comment (RC1)

The manuscript by Athanasiou et al. proposes a surrogate model of a time-consuming numerical model (XBeach) for the prediction of coastal erosion under extreme conditions at the regional scale (the Netherlands' coast). The main benefit of the approach is speeding up the prediction process without significantly decreasing the accuracy of erosion estimates. This is particularly useful in the application of early warning systems. Similar methodologies have been proposed before, but the novelty here is the application on a regional scale and the inclusion of the beach profile as an input. The latter makes the model capable of reproducing coastal erosion for a wide range of typological coast profiles, enabling accurate and fast predictions regardless of the initial state of the beach. The framework is presented as a flexible and transferable tool, so similar applications to other processes should be viable.

The manuscript is well presented, it reads well, and figures are of scientific rigor. The proposed methodology is thoroughly explained and easy to interpret, facilitating its future application to similar issues. There are some instances where the quality of the manuscript would benefit from clarifications. For example, a few (but important) details of the statistical model used here were not explained in detail. Correcting this will not require great effort, hence I suggest accepting the manuscript after minor revisions.

Please find below a more detailed review, providing line numbers where some corrections or clarifications might help improve the quality of manuscript. I hope you find my suggestions useful.

Kind regards.

Line 72. I would expect a sentence here highlighting the novelty and relevance of the proposed model as compared to the ones mentioned just before.

Line 125. What was the criteria for determining highly dynamic areas?

Line 142. There could be Hs extreme events coinciding with a non-extreme SSL (and vice versa). What if, for instance, SSL (even though it does not exceed a threshold) is high enough to cause erosion together with an extreme Hs. Wouldn't it be more appropriate to include extreme Hs AND/OR extreme SSL?

Line 145. Remove (2016)

Line 172. I wonder if a GPD is appropriate to describe the marginal of D. D is not defined based on threshold exceedances here, and although a GPD can be used to describe other extremes (such as annual maxima), I am not convinced this variable may have a heavy-tail behaviour.

Line 176. A Gaussian copula is mentioned here, so I assume it is a Multivariate Gaussian copula what is used here to model the dependence structure between the four hydrodynamic variables. This copula is chosen based on earlier work by Li et al. (2014b) as it was deemed suitable at the Ijmuiden-06 station. This may seem a reasonable choice, as long as the data used here does not differ substantially from Li et al. (2014b). But I wonder if the

Multivariate Gaussian copula would also be appropriate for the other stations. I can see a validation at the Euro platform station (Figure 3), but not for the other stations.

Line 180. 100,000 synthetic events are sampled based on Monte-Carlo. How does this number translate into length of data? What is the rate of events per year?

Line 185. Related to the previous comment in line 176, it is stated here that dependency statistics were smaller than 5%, but I have the feeling this only refers to the data presented in Figure 3. Could you also report metrics for the other stations?

Figure 3. The resolution of this figure should be increased. Also, how do the copula-based most extreme simulations presented here compare with estimates of most extreme historic events not included in the copula analysis as observations, such as the 1953 one? This could also give an indication of how realistic the most extreme synthetic events are, especially for simulations far more extreme than the observed ones. Perhaps a return period/value comparison between extreme historical events not included as observations (e.g., 1953) and synthetic events from the copula analysis would be insightful and reinforce your message about the suitability of the statistical model.

Line 275. You may mention this later, but how was this division done? 50/50? Was this a k-fold validation? How did you determine a suitable calibration/dataset division to ensure that was a good way of selecting the architecture of the ANNs?

Figure 10. Encouraging that the model performs best for the most complete pre- and poststorm profiles available (2019), but it is also true that this event was not particularly erosive. Would the model perform as well for more erosive events if we had complete prepost/storm profiles as in 2019? I seem to remember the winter season of 2013-14 was particularly extreme (waves and surge) in northern Europe (this especially applied to the UK, but I imagine the Netherlands was also impacted by these series of storms). Are there records of complete pre- post-storm transects for that particularly extreme winter? It could be more insightful to show how the model performs for more erosive events while being validated with complete transects (if they are available).

Figure 12. This is a nice and interesting figure. I wonder what's the effect of altering the number of TCPs included in the training process.

Line 552. There could be also problems in tropical-storm prone areas. Especially the fitting of marginal distributions and the copula approach, given the rarity of those events.

---

## Author Comment (AC1)

We would like to thank dr. Víctor Malagón-Santos (Reviewer 1) for his thorough review and constructive suggestions. In this response we are addressing his comments and feedback. We have numbered the reviewer's comments (**R1.1, R1.2** etc.) in order to facilitate referencing to each comment. We have added here all the changes in manuscript (shown with green). The pages and line numbers in our responses refer to the revised manuscript with track changes.

**Reviewer 1 (Víctor Malagón-Santos)**

**The manuscript by Athanasiou et al. proposes a surrogate model of a time-consuming numerical model (XBeach) for the prediction of coastal erosion under extreme conditions at the regional scale (the Netherlands' coast). The main benefit of the approach is speeding up the prediction process without significantly decreasing the accuracy of erosion estimates. This is particularly useful in the application of early warning systems. Similar methodologies have been proposed before, but the novelty here is the application on a regional scale and the inclusion of the beach profile as an input. The latter makes the model capable of reproducing coastal erosion for a wide range of typological coast profiles, enabling accurate and fast predictions regardless of the initial state of the beach. The framework is presented as a flexible and transferable tool, so similar applications to other processes should be viable.**

**The manuscript is well presented, it reads well, and figures are of scientific rigor. The proposed methodology is thoroughly explained and easy to interpret, facilitating its future application to similar issues. There are some instances where the quality of the manuscript would benefit from clarifications. For example, a few (but important) details of the statistical model used here were not explained in detail. Correcting this will not require great effort, hence I suggest accepting the manuscript after minor revisions.**

We would like to thank the Reviewer for the positive comments.

**Please find below a more detailed review, providing line numbers where some corrections or clarifications might help improve the quality of manuscript. I hope you find my suggestions useful.**

**R1.1:**
**Line 72. I would expect a sentence here highlighting the novelty and relevance of the proposed model as compared to the ones mentioned just before.**

We agree with the reviewer and have now added this part:

[Line 72]: "The novelty of the proposed meta-model relative to previous similar methodologies is the inclusion of the pre-storm beach profile as input, which allows for large scale applications."

**R1.2:**
**Line 125. What was the criteria for determining highly dynamic areas?**

The highly dynamic areas included areas like the sand spits formed at the tips of the Wadden islands, were either way no dune features are present. This included some areas on the back of the Wadden islands (which are protected from North Sea storms). We have now changed the text to:

[Line 126]: "We excluded some of the 1,430 profiles from Athanasiou et al. (2021), which were found to be at highly dynamic areas (e.g., transect at the tips of the Wadden islands, where no clear dune features were identified) or transects at the non-exposed side of the Wadden islands, leaving 1,368 transect for our study."

**R1.3:**
**Line 142. There could be Hs extreme events coinciding with a non-extreme SSL (and vice versa). What if, for instance, SSL (even though it does not exceed a threshold) is high enough to cause erosion together with an extreme Hs. Wouldn't it be more appropriate to include extreme Hs AND/OR extreme SSL?**

While swell waves can occur at the Dutch coast even when SSL is not extreme, we don't expect these events to be impactful due to the high dune elevation at the Dutch coast. Furthermore, we picked these thresholds per station, based on the morphological characteristics of each region (see Lines 147-155), which means that they are representative of what constitute an event (i.e., collision regime) based on the local dune toe elevation.

**R1.4:**
**Line 145. Remove (2016)**

We have now removed it.

**R1.5:**
**Line 172. I wonder if a GPD is appropriate to describe the marginal of D. D is not defined based on threshold exceedances here, and although a GPD can be used to describe other extremes (such as annual maxima), I am not convinced this variable may have a heavy-tail behaviour.**

We based this choice on the results of Li et al. (2014), where a GPD was used to model the tail of event duration as well. To validate this, we performed a Kolmogorov–Smirnov test for all GPD fits and they were passed at the 95% confidence interval.

Li, F., Van Gelder, P. H. A. J. M. , Vrijling, J. K., Callaghan, D. P., Jongejan, R. B., & Ranasinghe, R. (2014). Probabilistic estimation of coastal dune erosion and recession by statistical simulation of storm events. Applied Ocean Research, 47, 53–62. https://doi.org/10.1016/j.apor.2014.01.002

**R1.6:**
**Line 176. A Gaussian copula is mentioned here, so I assume it is a Multivariate Gaussian copula what is used here to model the dependence structure between the four hydrodynamic variables. This copula is chosen based on earlier work by Li et al. (2014b) as it was deemed suitable at the Ijmuiden-06 station. This may seem a reasonable choice, as long as the data used here does not differ substantially from Li et al. (2014b). But I wonder if the Multivariate Gaussian copula would also be appropriate for the other stations. I can see a validation at the Euro platform station (Figure 3), but not for the other stations.**

Indeed, this is refereeing to a multivariate Gaussian copula. We have now added this to the text to make it more clear:

[Line 181]: "...and we then fit a multivariate Gaussian copula..."

We had decided not to include the pair-plots for each station in the initial manuscript for the sake of space. But the validation was done for all stations. We have changed this part of the text:

[Line 186]: "The relative differences between the observed and simulated dependency statistics

were on average smaller than 5%, 1% and 2% for ρ, τ, and TDC respectively, for all four stations (Figure 3 and Supplementary Figures S1-S3), verifying that the simulator was able to capture the dependency structure."

Additionally, now we have added a supplement presenting the event simulation and validation statistics for the other stations:

[Figure]

**Figure S1: Copula-based events simulator for the Schiermonnikoog Noord station location (see Figure 2). Red, grey and black dots indicate observed, simulated and the 100 MDA selected events, respectively. The black dots are a subset of the grey dots (simulated events) that are selected with the MDA. Histograms of each storm parameter (*Hs, Tp, SSL* and *D*) for both observed and simulated events can be seen in the diagonal graphs. Below the diagonal, scatter-plots for each pair are plotted. Above the diagonal, 3 different dependency coefficients (ρ: Pearson correlation, τ: Kendall's rank correlation, TDC: non-parametric tail dependence) are shown for each pair for the observed and simulated events.**

[Figure]

**Figure S2: Copula-based events simulator for the Eierlandse Gat station location (see Figure 2). Red, grey and black dots indicate observed, simulated and the 100 MDA selected events, respectively. The black dots are a subset of the grey dots (simulated events) that are selected with the MDA. Histograms of each storm parameter (*Hs, Tp, SSL* and *D*) for both observed and simulated events can be seen in the diagonal graphs. Below the diagonal, scatter-plots for each pair are plotted. Above the diagonal, 3 different dependency coefficients (ρ: Pearson correlation, τ: Kendall's rank correlation, TDC: non-parametric tail dependence) are shown for each pair for the observed and simulated events.**

[Figure]

**Figure S3: Copula-based events simulator for the IJmuiden-06 station location (see Figure 2). Red, grey and black dots indicate observed, simulated and the 100 MDA selected events, respectively. The black dots are a subset of the grey dots (simulated events) that are selected with the MDA. The purple star indicates the boundary conditions during the 1953 storm. Histograms of each storm parameter (*Hs, Tp, SSL* and *D*) for both observed and simulated events can be seen in the diagonal graphs. Below the diagonal, scatter-plots for each pair are plotted. Above the diagonal, 3 different dependency coefficients (ρ: Pearson correlation, τ: Kendall's rank correlation, TDC: non-parametric tail dependence) are shown for each pair for the observed and simulated events.**

**R1.7:**
**Line 180. 100,000 synthetic events are sampled based on Monte-Carlo. How does this number translate into length of data? What is the rate of events per year?**

Since the objective of this step for this research was to create a representative set of training storms and not derive probabilistic risk estimates, the number of simulated events was chosen high enough to make sure that extremes are sampled well. But we did not model the frequency of occurrence of the events in a year. From the observed events though, the average number of events per year was ~6-7. Which would mean that 100,000 synthetic events would represent ~14,000 years, but as mentioned before we don't use this in our analysis.

**R1.8:**

**Line 185. Related to the previous comment in line 176, it is stated here that dependency statistics were smaller than 5%, but I have the feeling this only refers to the data presented in Figure 3. Could you also report metrics for the other stations?**

Please see our response to R1.6. We have now explicitly mention that this refers to all stations and have now added the figures of the other stations in a supplement.

**R1.9:**

**Figure 3. The resolution of this figure should be increased. Also, how do the copula-based most extreme simulations presented here compare with estimates of most extreme historic events not included in the copula analysis as observations, such as the 1953 one? This could also give an indication of how realistic the most extreme synthetic events are, especially for simulations far more extreme than the observed ones. Perhaps a return period/value comparison between extreme historical events not included as observations (e.g., 1953) and synthetic events from the copula analysis would be insightful and reinforce your message about the suitability of the statistical model.**

Any issues with the current resolution probably relate to the pdf conversion during the submission. During the final submission all figures will be provided in high resolution.

Following the observation from the Reviewer, we re-evaluated available data on the boundary conditions of the 1953 storm, and found that for the closest station of Hoek van Holland, a max TWL of ~ 3.85 m was observed. Using the local MHW from the Jarkus dataset this leads to a SSL of 2.84 instead of the 3 m we previously used. We have now plotted this event in the pair plots of the IJmuiden-06 station. It can be seen that for the SSL the 1953 is slightly larger than the largest SSL simulated. This can be connected with the difference in the way we calculated SSL for the observed events (we did a tidal analysis to calculate the historic tides from the TWL record), while for the 1953 storm we simply subtract the local MHW from the maximum TWL record. For the other parameters, ($H_s$, $T_p$ and D) the 1953 storm fall well inside the simulated events cloud.

[Figure]

**Figure S4: Copula-based events simulator for the IJmuiden-06 station location (see Figure 2). Red, grey and black dots indicate observed, simulated and the 100 MDA selected events, respectively. The black dots are a subset of the grey dots (simulated events) that are selected with the MDA. The purple star indicates the boundary conditions during the 1953 storm. Histograms of each storm parameter (*Hs, Tp, SSL* and *D*) for both observed and simulated events can be seen in the diagonal graphs. Below the diagonal, scatter-plots for each pair are plotted. Above the diagonal, 3 different dependency coefficients (ρ: Pearson correlation, τ: Kendall's rank correlation, TDC: non-parametric tail dependence) are shown for each pair for the observed and simulated events.**

Using this updated boundary condition for SSL, we redone the validation of section 3.3. We the updated values, the predicted DEV for the 1953 is actually closer to the observed values than before.

Table 2:

| Storm | SSL (m) | Hs (m) | Tp (s) | D (h) | Number of transects | Remarks |
|---|---|---|---|---|---|---|
| February 1953 | 2.84 | 7.3 | 14.1 | 37 | 1 | A typical schematized profile of the Holland coast is used as the pre-storm profile, while for the observed dune erosion, reported values of 90±26 m³/m are used (Van Thiel de Vries, 2009). |

[Figure]

**Figure 5: Scatter plot between the observed (x-axis) and ANN-predicted (y-axis) DEV, for three historic storms with variable number of transects per storm. The dots indicate the average predictions of the ANN, while the vertical lines show the min and max values as given by the ensemble of the ANN output. For the storm of 1953 the horizontal line gives the range of the values reported.**

**R1.10:**
**Line 275. You may mention this later, but how was this division done? 50/50? Was this a k-fold validation? How did you determine a suitable calibration/dataset division to ensure that was a good way of selecting the architecture of the ANNs?**

Indeed, it was not a k-fold validation, since the training dataset was already created in a "clever" way in order to capture representative profile and storm conditions. The benchmark dataset was sampled out of the training dataset to ensure that these cases are unseen to the model. In order to ensure that the results were not biased to a single split, we repeated the split 10 times with different randomizations seeds. We explain this in the text:

[Line 285]: "The division of the benchmark dataset to a calibration and validation dataset was performed 10 times with different randomization seeds and the final mean error statistics were used, to ensure that any bias of the individual divisions was minimized. This meant that 10 different ANNs were produced (with the same architecture but different weights and biases), which will give an ensemble of DEV predictions, that can work as an uncertainty range."

**R1.11:**
**Figure 10. Encouraging that the model performs best for the most complete pre- and post-storm profiles available (2019), but it is also true that this event was not particularly erosive. Would the model perform as well for more erosive events if we had complete pre- post/storm profiles as in 2019? I seem to remember the winter season of 2013-14 was particularly extreme (waves and surge) in northern Europe (this especially applied to the UK, but I imagine the Netherlands was also impacted by these series of storms). Are there records of complete pre- post-storm transects for that particularly extreme winter? It could be more insightful to show how the model performs for more erosive events while being validated with complete transects (if they are available).**

This is indeed a great observation. Sadly, there are no pre- and post-storm measurements available for the winter of 2013-2014 to perform this kind of validation. Only a small part (~250 m) of the

coast at Egmond aan Zee was measured before the storm season (Ruessink et al. 2019), and the post-storm measurements have the effects of two consecutive storms (October and December 2013), which do not allow for validation.

Nevertheless, even with the incomplete measurements of the storms of 1976 and 2019, we could see that the expected variability between the storms, and between the profiles was captured in an acceptable manner.

Ruessink, G., Schwarz, C. S., Price, T. D. and Donker, J. J. A.: A multi-year data set of beach-foredune topography and environmental forcing conditions at Egmond aan Zee, the Netherlands, Data, 4(2), http://doi.org/10.3390/data4020073, 2019.

**R1.12:**
**Figure 12. This is a nice and interesting figure. I wonder what's the effect of altering the number of TCPs included in the training process.**

While this is a nice suggestion, it is not straight forward to compare this since in contradiction to the MDA algorithm that was used for the storms, the K-Means algorithm which was used for the TCPs does not maintain the order or the actual TCPs that are chosen for each cluster. This would mean that we would have to repeat all XBeach simulations for each scenario with a different number of TCPs, which was deemed out of the scope of the present study.

Nevertheless, we have already studied this effect in a previous paper focusing on clustering elevation profiles at the Dutch coasts (Athanasiou et al. 2021). There we tested different numbers of TCPs and saw that 100 TCPs was optimum to balance the computational effort and the good representation of the coast.

**R1.13:**
**Line 552. There could be also problems in tropical-storm prone areas. Especially the fitting of marginal distributions and the copula approach, given the rarity of those events.**

**We have now added this part in the text:**

[Line 554]: "Additionally, special care should be taken in applying these methods at tropical-storm prone areas, where it should be ensured that the training forcing conditions are representative of the extremes induced by tropical cyclones (Bloemendaal et al., 2022)."

---

## Author Comment (AC2)

We would like to thank Reviewer 2 for his/her thorough review and constructive suggestions. In this response we are addressing his comments and feedback. We have numbered the reviewer's comments (**R2.1, R2.2** etc.) in order to facilitate referencing to each comment. We have added here all the changes in manuscript (shown with green). The pages and line numbers in our responses refer to the revised manuscript with track changes.

**Reviewer 2**

**Athanasiou et al. present a framework that combines artificial neural networks and process-based modelling (XBeach) to derive estimates of dune erosion during storms on the entire Dutch coast. The approach yields a prediction skill with an RMSE of 19 m3/m, which is reasonable given the 1D-approach and the simplification of the hydrodynamic boundary conditions. The model can provide estimates of dune erosion volumes within seconds, making it a crucial new approach for assessing potential dune erosion hot spots as storms develop and hit the coast.**

**The authors provide a very well-written and clearly structured manuscript. The relevance and need for their work is clearly outlined in the introduction section, and reflected upon in the discussion section. The approach is detailed in an elaborate manner, making it reproducible for application of the technique elsewhere. Assumptions and limitations underlying the approach are made explicit and tested as part of the results. There is room for improvement, which is clearly addressed by the authors and this work forms a solid basis to do so. The inclusion of oblique wave incidence would be a crucial next step. As such, this work is relevant to the research field and the readers of NHESS and, after taking the comments below into account, I recommend this manuscript to be published with minor revisions.**

We would like to thank the Reviewer for the positive comments.

**COMMENTS**

**R2.1:**
**L84-86 A preview into the method is given here, by introducing the two-step approach with (1) a classifier and (2) a regressor. This is further elaborated in the methods section. It is unclear to me why the first step is needed if the regressor may also yield DEV=0. Please elaborate on why this choice was made.**

The regressor cannot yield DEV=0, since the DEV prediction is transformed from the logarithmic scale (see Line 319: "*...and the final DEV estimation was acquired by using the exponential function on the output neuron to transform them back to real units.*") and trained only with cases that have DEV > 0 (see Lines 316-317: "*For the regression ANN only the cases where DEV > 0 were used, which comprised 79% and 77% of the total cases in the training and benchmark cases, respectively.*").

The choice of having both a classifier and a regressor had a two-fold purpose. First, after some testing we found that the classifier by itself had a better skill in estimating the binary dune erosion response, i.e. there is or there is not a dune erosion event, when compared to using a regressor that would have to estimate erosion quantities as well as non-erosion events. Secondly, the exclusion of cases with DEV<=0, during the training of the regressor, allowed for a log-transform of the DEV quantities in the training of the regression ANN, which resulted in a better model performance during prediction at different orders of magnitude of DEV (see Lines 317-318: "*The DEV values were first transformed to log(DEV) to ensure that during the calculation of losses, similar importance was given to minor and larger DEV cases (van Gent et al., 2007).*").

We summarize these points in:

[Lines 272-273] : "With this 2-phase approach we ensured that overprediction for smaller DEVs was avoided and enabled the prediction of zero DEV (Verhaeghe et al., 2008)."

**R2.2:**
**L227 Here the reader is referred to the discussion section on the impact of assuming shore normal wave-incidence for dune erosion predictions; this may lead to an underprediction of dune volumes. Please briefly mention (1-2 sentences) the implications of this here, in section 2.4, so it is clear to the reader before starting to interpret the results.**

We agree and we have now added this sentence:

[Lines 229-231]: "This assumption may lead to underestimation of dune erosion when the generation of alongshore currents becomes important for sediment stirring and thus sediment transport (de Winter and Ruessink, 2017)."

**R2.3:**
**L233-235 DEV<0 was predicted by XBeach, and the presence of newly-formed dunes is mentioned as a possible cause and deemed non-representative of the dune response. Such local accretion has been observed elsewhere, e.g. Cohn et al, GRL (2018) or Harley et al., Nature (2022), and may result from alongshore variability in pre-storm morphology. This cannot be accounted for by the 1D model used in this study, but such variations may develop during relatively small SSL, as pointed out by the authors. I feel this should be mentioned for completeness and for the translation of the model results to field observations.**

This is a very relevant comment and we thank the reviewer for the references. We have now added this sentence:

[Lines 241-243]: "Additionally, local accretion can be connected with alongshore variability of pre-storm morphology (Cohn et al., 2018; Harley et al., 2022), but cannot be resolved with the 1D approach used in this study."

**R2.4:**
**Section 4.2**
**I agree that including oblique wave angles would be an important next step, as storms on the Dutch are not shore-normally incident everywhere along the coast.**

We agree and added this:

[Line 503]: ".. storms are not everywhere shore-normally incident along the Dutch coast ..."

**R2.5:**
**Would this model be able to capture the response (DEV) to a storm of a dune that has not yet recovered from a previous storm? Or is this captured in the morphological inputs fed to the ANN? I.e. can this model structure deal with sequences of storms?**

This is a really interesting point raised by the reviewer. In its current format the meta-model cannot directly deal with storm sequences, since the output is an indicator (DEV) and not the complete

post-storm profile. Information on the new profile morphological characteristics after the 1st storm would be needed as input to make an estimate for the 2nd storm. To this end, this could be accomplished if the meta-model is updated to extract post-storm profiles (or profiles characteristics) instead of only an erosion indicator.

**R2.6:**
**In addition to the parameters mentioned that may also be of interest and are reported in the literature, the authors may consider adding nearshore bar morphology, as observed by e.g. Castelle et al 2015 and touched upon by the authors in the introduction section. This also relates to the possible future expansion of the method discussed in L540-544.**

We have now added this part:

Lines [520-521]: "In the presented meta-model, the nearshore area of the profile was only described by the nearshore slope. However, pre-storm bar morphology is of importance for post-storm dune erosion (Castelle et al., 2015)."

**R2.7:**
**L549-550 The presented framework would be a very useful starting point for application elsewhere. Will the code be made openly available (e.g. a Jupiter notebook, or through GitHub)? If so, please include the link.**

The ANN framework was developed based on the Python packages Keras and Tensorflow, which are free to use. In essence, for an application elsewhere, these packages could be employed with a new training dataset derived for the case study. To this end, it was decided not to make the code openly available, since the manuscript outlines in detail all the steps that are needed for an application elsewhere. Nevertheless, we would mention in the manuscript that the code can be provided upon request.